# Perineural Invasion Predicts Local Recurrence and Poor Survival in Laryngeal Cancer

**DOI:** 10.3390/jcm12020449

**Published:** 2023-01-05

**Authors:** Hyun-Il Shin, Joo-In Bang, Geun-Jeon Kim, Dong-Il Sun, Sang-Yeon Kim

**Affiliations:** Department of Otorhinolaryngology and Head & Neck Surgery, College of Medicine, The Catholic University of Korea, Seoul St. Mary’s Hospital, Seoul 06591, Republic of Korea

**Keywords:** laryngeal cancer, laryngectomy, perineural invasion, prognosis, recurrence

## Abstract

(1) Background: Perineural invasion (PNI) in head and neck cancer is associated with a poor prognosis; however, the effect of PNI on the prognosis of laryngeal cancer remains under debate. This retrospective study aimed to investigate the effect of PNI in fresh or salvaged larynges on survival in patients who had undergone laryngectomy for squamous cell carcinoma. (2) Methods: This study enrolled 240 patients diagnosed with laryngeal cancer who had undergone open surgery at Seoul St. Mary’s Hospital, Korea. The effects of PNI, other histopathologic factors, and treatment history on survival and recurrence patterns were assessed. (3) Results: PNI was observed in 30 of 240 patients (12.5%). PNI (HR: 3.05; 95% CI: 1.90–4.88; *p* = 0.01) was a significant predictor of poor 5-year disease-free survival. In fresh cases, preepiglottic invasion (HR: 2.37; 95% CI: 1.45–3.88; *p* = 0.01) and PNI (HR: 2.96; 95% CI: 1.62–2.96; *p* = 0.01) were negative prognostic factors for 5-year disease-free survival. In the salvage group, however, only PNI (HR: 2.74; 95% CI: 1.26–5.92; *p* = 0.01) was a significant predictor of disease-free survival. Further, PNI significantly influenced high local recurrence (HR: 5.02, 95% CI: 1.28–9.66; *p* = 0.02). (4) Conclusions: Independent of treatment history, PNI is a prognostic factor for poor survival and local recurrence in laryngeal cancer.

## 1. Introduction

Laryngeal cancer is the second most common malignancy of the upper digestive tract, with squamous cell carcinoma constituting the majority of tumors [1,2,3]. Multiple factors affect the prognosis of patients with laryngeal cancer, including stage, surgical margin status, tumor differentiation; the presence of extranodal extension (ENE), lymphovascular invasion, or distant metastases; and receipt of adjuvant therapy [4,5]. Among these factors, perineural invasion (PNI) is correlated with poor clinical outcomes in many cancers [5]. PNI was first defined by Batsakis in 1985 as “invasion in, around, and through” the nerves [6]. The most commonly accepted definition of PNI is “tumor in close proximity to the nerve and involving at least 33% of its circumference or tumor cells within any of the three layers of the nerve sheath” [5]. Head and neck cancers in particular have a high prevalence of PNI; its incidence is reported to vary between 6 and 80% [5,7,8,9,10,11,12,13]. Furthermore, PNI is a predictor of poor survival in head and neck cancers and is associated with high recurrence rates [7,14]. Therefore, PNI is categorized as a pathologic adverse feature in these cancers. If PNI is identified after surgery, adjuvant radiation therapy around the surgical bed is recommended. However, several studies have reported that PNI is not significantly correlated with prognosis, and few studies have been conducted on the relationship between PNI and prognosis in laryngeal cancer [15,16]. Consequently, evaluating the impact of pathologic factors, especially PNI, on the prognosis of laryngeal cancer is warranted. Therefore, we designed this study to (1) investigate the incidence of PNI and other histopathologic factors in fresh or salvaged larynges from patients who had undergone laryngectomy for squamous cell carcinoma, and (2) study the impact of these predictors on survival and recurrence patterns according to treatment modality and PNI status in laryngeal cancer.

## 2. Materials and Methods

### 2.1. Patient Selection

We performed a retrospective chart review of 240 patients diagnosed with squamous cell carcinoma of the larynx who had undergone open surgical resection of the primary tumor with neck dissection at the Department of Otolaryngology-Head and Neck Surgery, Seoul St. Mary’s Hospital, Catholic University of Korea (Seoul, Korea), between January 2000 and January 2021. The treatment protocol at this center was determined by the tumor board and was based on the National Comprehensive Cancer Network (NCCN) guidelines. Patient characteristics, including sex, age, and smoking status, were analyzed. Patients were divided into three groups based on smoking status: (1) never smoked; (2) smoked > 40 pack years; and (3) smoked < 40 pack years. Patients were assigned tumor (T)- and node (N)-classifications corresponding to the eighth edition of the American Joint Committee on Cancer tumor-node-metastasis staging system. Excision of the laryngeal tumor site was performed using open-framework surgery, and patients treated via a transoral approach were excluded. Modified radical neck dissection was performed on the involved side in patients with clinically suspected or cytologically proven positive lateral lymph nodes, whereas selective neck dissection was performed on those with clinical N0. The postoperative surveillance protocol consisted of chest and abdomen-pelvis computed tomography (CT), a bone scan, and primary-site magnetic resonance imaging every 6 months for 5 years after the end of treatment. If recurrence was suspected, additional tests including biopsy or positron emission tomography-CT, were performed. The time to recurrence was calculated from the day of surgery until recurrence was identified. For the risk factor analysis, the patients were divided into fresh-case (no prior treatment) and salvage groups. For each group, primary-site surgical specimens were carefully examined with respect to size, depth of invasion, margin involvement (clear: >5 mm, close: 1–5 mm, positive: <1 mm), paraglottic and preepiglottic space invasion, lymphatic and vascular invasion, and PNI.

### 2.2. Study Design

We compared survival outcomes and clinical and pathologic data of 240 patients who had undergone surgery for laryngeal cancer. Survival analysis was performed according to the presence of PNI in postoperative pathologic findings, other pathologic parameters, and previous treatment history. To review the causes of treatment failure, we investigated the pattern of recurrence (local recurrence, regional recurrence, or distant metastasis) according to the presence of PNI.

### 2.3. Statistical Analysis

Statistical analyses were performed using the Statistical Package for Social Sciences (SPSS 28; IBM Corp., Armonk, NY, USA). A student’s *t*-test and Pearson’s chi-square test were used to compare clinical demographics of patients and tumor characteristics in the overall population. Cox proportional hazards regression was performed for survival analysis. Statistical significance was set as *p* < 0.05, and results are reported as mean ± standard deviation.

## 3. Results

### 3.1. Patient Characteristics and Surgical Parameters

Among the 240 patients (90.1% male), the average age was 64.8 years (range: 42–89 years). A total of 150 patients (62.5%) had advanced T stages (T3 and 4), and 57 individuals (19.7%) had advanced N stages (N2 and 3). Supracricoid partial laryngectomy (43.7%) was the most common surgery type, followed by total laryngectomy (33.7%), supraglottic partial laryngectomy (13.7%), and vertical partial hemilaryngectomy (8.7%). Adjuvant radiation therapy only was administered to 49 patients (20.4%), while 48 (20.0%) patients received concurrent chemoradiation therapy. The primary site was the glottis in 66 (27.5%), supraglottis in 64 (26.6%), and transglottis in 110 (45.8%) enrolled patients. Of the 240 patients, 175 (72.9%) received surgery as primary treatment (fresh-case group), whereas 38 (15.8%) and 27 (11.2%) patients were previous radiation and surgery failure cases, respectively (salvage group) (Table 1).

### 3.2. Tumor Characteristics

Histopathologic features of tumors are summarized in Table 2. Well-differentiated tumors were observed in 66 patients (27.5%), while 18 patients (7.5%) showed poorly differentiated tissue. The average depth of invasion of the primary tumor was 10.37 ± 7.12 mm. Additionally, the average longest dimension of the primary tumor was 25.96 ± 13.34 mm. Regarding margin status, 54 patients (22.5%) exhibited a close margin, and 149 patients (62%) had negative margins. A total of 117 (48.7%) patients showed paraglottic invasion, and 43 (17.9%) patients had preepiglottic invasion. Furthermore, lymphatic and vascular invasion was observed in 60 (25.0%) and 13 patients (5.4%), respectively. PNI was identified in 30 patients (12.5%).

### 3.3. Risk Factors for 5-Year Overall Survival

Cox regression analysis was performed to determine the risk factors for 5-year overall survival (OS) of the patients according to previous treatment modality. In the entire cohort, paraglottic (hazard ratio [HR]: 1.49; 95% confidence interval [CI]: 1.04–2.14; *p* = 0.02) and preepiglottic space invasion (HR: 1.87; 95% CI: 1.21–2.88; *p* = 0.04), lymphatic invasion (HR: 1.80; 95% CI: 1.23–2.62; *p* = 0.02), and PNI (HR: 3.19; 95% CI: 1.99–5.12; *p* = 0.01) significantly influenced the 5-year OS. In fresh cases specifically, preepiglottic space invasion (HR: 2.23; 95% CI: 1.36–3.66; *p* = 0.01), lymphatic invasion (HR: 1.67; 95% CI: 1.04–2.68; *p* = 0.03), and PNI (HR: 2.73; 95% CI: 1.47–5.09; *p* = 0.01) were statistically significant prognostic factors for OS. Conversely, only PNI (HR: 2.80; 95% CI: 1.28–6.12; *p* = 0.01) was a statistically significant survival predictor in the salvage group (Table 3).

### 3.4. Risk Factors for 5-Year Disease-Free Survival

We further investigated the risk factors for 5-year disease-free survival (DFS) according to treatment status. Similarly to the OS findings, preepiglottic space invasion (HR: 1.92; 95% CI: 1.25–2.96; *p* = 0.03), lymphatic invasion (HR: 1.74; 95% CI: 1.19–2.54; *p* = 0.04), and PNI (HR: 3.05; 95% CI: 1.90–4.88; *p* = 0.01) were significant factors for 5-year DFS in the entire cohort. In the fresh-case group, preepiglottic space invasion (HR: 2.37; 95% CI: 1.45–3.88; *p* = 0.01) and PNI (HR: 2.96; 95% CI: 1.62–2.96; *p* = 0.01) were significant prognostic factors, whereas only PNI (HR: 2.74; 95% CI: 1.26–5.92; *p* = 0.01) was a statistically significant prognostic factor for DFS in the salvage group (Table 4).

### 3.5. Patterns of Recurrence According to PNI Status

During the follow-up period, disease recurrence was confirmed in 44 patients. We analyzed the pattern of recurrence in relation to PNI status. The overall recurrence rate in the PNI-positive patient group was significantly higher than that in the PNI-negative group (26.7% vs. 17.1%, HR: 3.06, 95% CI: 1.57–6.00; *p* = 0.04). Moreover, the local recurrence rate specifically was significantly higher in PNI-positive than in PNI-negative patients (10% vs. 3.8%, HR: 5.02, 95% CI: 1.28–9.66; *p* = 0.02) (Table 5).

### 3.6. Survival Analysis According to PNI Status and Treatment History

The median follow-up time was 65.65 months (range: 18–244 months). In the entire cohort, the 5-year OS rate was 66.7% for PNI-negative and 29.3% for PNI-positive patients (*p <* 0.001). In the fresh-case group, the 5-year OS rates were 71.5% and 31.1% in the PNI-negative and -positive groups, respectively (*p <* 0.001). Further, in salvage patients, the 5-year OS rates of PNI-negative and -positive patients were 52.3% and 31.7%, respectively (*p =* 0.007) (Figure 1). The 5-year DFS rate was 65.6% in the PNI-negative and 28.9% in the PNI-positive groups in the entire cohort (*p <* 0.001). In the fresh-case and salvage groups, the DFS rates were 70.0% (PNI-positive) and 31.1% (PNI-negative; *p <* 0.001) and 52.5% (PNI-negative) and 30.8% (PNI-positive; *p =* 0.007), respectively (Figure 2).

## 4. Discussion

In the last few decades, the survival rate for laryngeal cancer has improved with the use of organ-preservation treatment regimens; however, 22–31% of patients with laryngeal cancer develop recurrence 2–3 years after treatment termination [17,18]. Despite numerous therapeutic and histopathologic studies, no morphological markers are currently available to predict survival outcomes in patients with laryngeal cancer. According to the NCCN guidelines (version 2.2022), several tumor-related characteristics are associated with a higher chance of recurrence in laryngeal cancer, namely advanced T and/or N stage, positive (<1 mm) or close resection margin (1–5 mm), PNI, vascular and/or lymphatic invasion, and ENE. Adjuvant treatment is recommended in these cases. Other alternative factors, including age [19] and tumor differentiation [20,21], have also shown prognostic significance.

In many previous studies, PNI has been reported as a strong prognostic marker that indicates the aggressive behavior of laryngeal cancer [8,14,22]. Although first described in the 19th century, the early works of Ballantune et al. [23] and Batsakis [6] in the 20th century brought greater clinical awareness to the significance of PNI in laryngeal cancer. Batsakis’s definition of PNI in 1985 characterized it as tumor cell invasion in, around, and through nerves, a broad category that encompasses most observations. In their review, Liebig et al. [5] advocate for defining PNI as “the finding of tumor cells within any of the three layers of the nerve sheath.” PNI has been reported across many head and neck mucosal squamous cell carcinoma case series at prevalence rates between 25% and 80% [24,25,26,27]. Recently, in addition to laryngeal and head and neck cancers, PNI has been recognized as a significant predictive factor for poor prognosis in various solid cancers, including those of the pancreas, colon and rectum, prostate, biliary tract, and stomach [28,29,30,31,32].

PNI is the process of neoplastic invasion of nerves; to understand the clinical implications of PNI, one must first understand the anatomical structure of the nerve. The nerve sheath is composed of three layers: the epineurium, perineurium, and endoneurium. The outer layer, the epineurium, presents a rich vascular network (vasa nervorum) and lymphatic vessels surrounding the loose areolar connective tissue; inside is a dense structure made of collagen and elastin fibers. For many years, the lymphatic vessels of the epineurium were believed to represent the path for neoplastic spread. Maddox [33] and McGavran et al. [34] reported that PNI is an independent predictor of cervical lymph node metastases; therefore, patients with PNI-positive cancers should undergo elective neck treatment. Furthermore, according to Brown et al. [35], PNI is associated with both an increased risk of cervical metastases and decreased survival rates, because a tumor that involves the perineural space can spread in both a longitudinal and radial fashion through the planes of least resistance. This increases the likelihood of the tumor extending beyond the margins of resection, increasing the incidence of local recurrence.

However, many recent studies have shown that the lymphatic vessels do not penetrate the inner, dense aspect of the epineurium [6,7,12,36]. These definitive studies have proven that there are no lymphatics within the inner nerve sheath and that several layers of collagen and basement membrane separate the inside of the nerve from the surrounding lesion; this is not a low-resistance path. Studies have demonstrated that PNI is a deliberate, molecularly mediated process that results from reciprocal interactions between cancer and nerves, challenging the historical notion that PNI is driven purely by the progression of cancer alone [37,38]. In an in vitro model, Ayala et al. [39] demonstrated tumor cell migration along neurites toward the ganglia of origin, as well as focused, directional outgrowth of neurites toward cancer cell colonies. The addition of stromal cells to the model increased neurite outgrowth and cell colony formation [40]. These findings suggest that the signaling mechanisms underlying PNI likely involve at least three different cellular elements, including tumor, nerve, and stromal cells, and may include autocrine and paracrine mechanisms. Axonal growth and neurite formation are complex processes that require neurotrophic factors, including nerve growth factor, brain-derived neurotrophic factor, neurotrophin 3 and 4/5, and axonal guidance molecules [41,42,43]. The upregulation not only of neurotrophins but also of factors such as matrix metalloproteinases (MMPs), particularly the gelatinases MMP-2 and -9, facilitates cancer cell migration through PNI at the stromal level [44,45].

In our study, PNI was identified as a poor prognostic factor for OS (HR: 3.19; 95% CI: 1.99–5.12; *p* = 0.01) and DFS (HR: 3.05; 95% CI: 1.90–4.88; *p* = 0.01) in laryngeal cancer. In addition, there was a significant difference in the recurrence rate according to PNI status; specifically, the local recurrence rate was significantly higher in PNI-positive patients (HR: 5.02, 95% CI: 1.28–9.66; *p* = 0.02). The mechanisms of perineural spread of tumors due to the interaction of cancer cells and nerves at various levels could explain these results. Although PNI is associated with poor prognosis, there are several characteristics that suggest the worst prognosis in head and neck cancer, including the extent of PNI involvement and number of foci, caliber of the largest involved nerve, presence of “skipping” involvement longitudinally along the nerve, intraneural invasion, intratumoral vs. extratumoral location of PNI, and involvement of a large-caliber or “named” nerves [7,27,46,47,48,49,50,51,52,53]. The independent impact of these histologic features is controversial, and further classification is needed to assess their precise clinical effects [52,54].

Because PNI is a negative prognostic factor, adjuvant radiation therapy is typically recommended for cases in which PNI is confirmed postoperatively. However, in recent studies [8,55], PNI-positive patients showed a poorer prognosis than PNI-negative patients despite postoperative radiotherapy. This result is consistent with the findings reported by Langendi et al. [56], who retrospectively analyzed 801 patients with head and neck cancer who had undergone postoperative radiotherapy. Additionally, a randomized trial identified a survival advantage for patients receiving cisplatin concurrently with postoperative radiation therapy compared to those receiving radiation therapy alone [57], suggesting that more aggressive postsurgical treatment may improve the prognosis in PNI-positive patients.

In this study, the impact of positive surgical margin on 5-year OS and DFS was not statistically demonstrated. In several previous studies, positive surgical margin has been known to adversely affect local control rate and survival of the disease [58,59]. Hinerman et al. [59] reported a 5-year locoregional control rate of 56% and 89% (*p* = 0.075) for negative and positive surgical margins in laryngeal cancer, respectively. However, the potential role of surgical resection margin is still being debated, since there are no definitive data available that take into account the heterogeneity of laryngeal cancer from a staging and biomolecular point of view. Furthermore, recent studies have shown that the positive resection margin does not significantly affect the prognosis in laryngeal cancer patients, and these results are presumed to be the influence of adjuvant chemotherapy and radiation therapy [60,61].

Our study has several limitations: (1) its retrospective nature, (2) a small sample size, (3) heterogeneity of the study population in clinical stage, treatment history, and postoperative adjuvant therapy, (4) a short-term follow-up period, and (5) lack of the result of multivariate analysis. Nevertheless, to the best of our knowledge, this is the largest study that has analyzed the association between laryngeal cancer prognosis and PNI. Furthermore, the cause of treatment failure was not clear in previous studies, but we concluded that PNI increased the risk of local failure and there was no significant correlation with regional failure and distant metastasis. We believe that this study has demonstrated the clinical significance of PNI in this disease, and that these findings may be helpful in determining adjuvant treatment policies for patients with advanced laryngeal cancer.

## 5. Conclusions

PNI-positive laryngeal cancer patients showed significantly worse 5-year OS and DFS compared with those of PNI-negative patients. Additionally, PNI was a significant negative prognostic marker for local recurrence after treatment termination. Although the current study had several limitations, the results may be used in the future as an important guide for determining treatment plans for patients with laryngeal cancer with PNI.

## Figures and Tables

**Figure 1 jcm-12-00449-f001:**
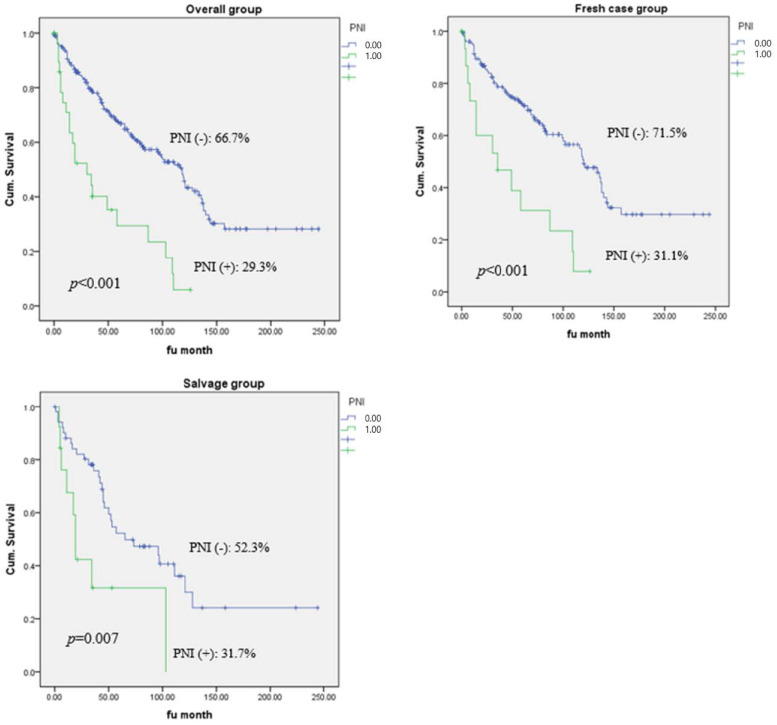
Five-year overall survival rate according to the status of perineural invasion.

**Figure 2 jcm-12-00449-f002:**
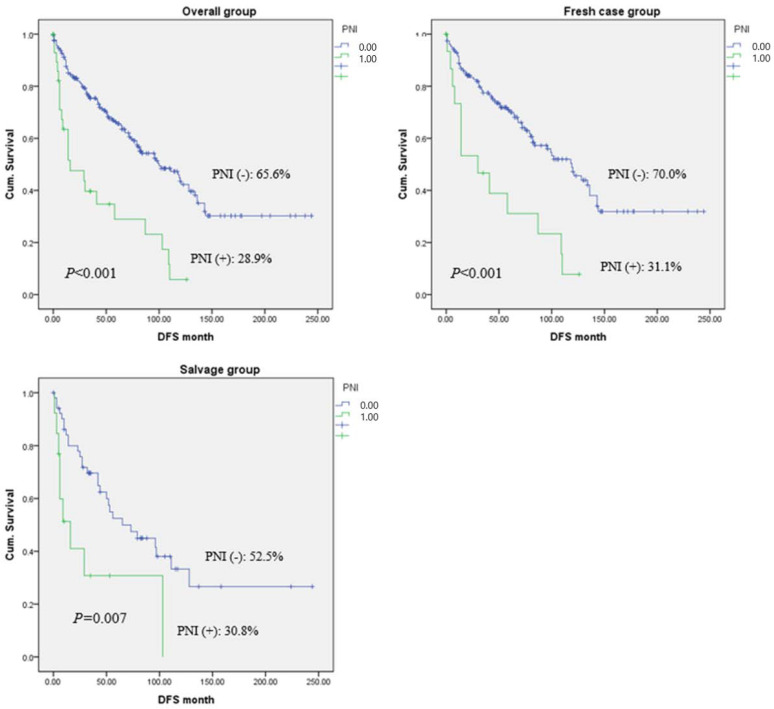
Five-year disease-free survival rate according to the status of perineural invasion.

**Table 1 jcm-12-00449-t001:** Patients characteristics.

Parameters	N (%)
**Sex** (male:female)	218:22
**Age** (years old, mean ± SD)	64.82 ± 9.63
**Smoking**	
40 pack years	97 (40.4%)
Less than 40 pack years	120 (50%)
**Primary Site**	
Glottis	66 (27.5%)
Supraglottis	64 (26.6%)
Transglottis	110 (45.8%)
**Previous treatment status**	
None (fresh case)	175 (72.9%)
RT failure	38 (15.8%)
Surgery failure	27 (11.2%)
**Type of surgery**	
Total laryngectomy	81 (33.7%)
Vertical partial hemilaryngectomy	21 (8.7%)
Supraglottic partial laryngectomy	33 (13.7%)
Supracricoid partial laryngectomy	105 (43.7%)
**TN stage (pathologic)**	
T1	23 (9.5%)
T2	67 (27.9%)
T3	84 (35%)
T4	66 (27.5%)
N0	165 (68.7%)
N1	18 (7.5%)
N2	36 (15%)
N3	21 (8.7%)
**Postoperative adjuvant therapy**	
RT	49 (20.4%)
CCRT	48 (20%)
None	143 (59.5%)

SD: Standard deviation, RT: Radiation therapy, CCRT: Concurrent chemo-radiation therapy.

**Table 2 jcm-12-00449-t002:** Histopathologic features of tumors.

Parameters	N (%)
Differentiation (1: WD, 2: PD)	66:18 (27.5%:7.5%)
Longest dimension (mm; mean ± SD)	25.96 ± 13.34
Depth of invasion (mm; mean ± SD)	10.37 ± 7.12
Margin status (clear:close:positive)	149:54:37 (62%:22.5%:15.4%)
Paraglottic space invasion	117 (48.7%)
Preepiglottic space invasion	43 (17.9%)
Lymphatic invasion	60 (25.0%)
Vascular invasion	13 (5.4%)
Perineural invasion	30 (12.5%)

WD: well differentiated, PD: poorly differentiated.

**Table 3 jcm-12-00449-t003:** Risk factor analysis for 5-year overall survival.

Parameters	Fresh Cases (*n* = 175)	Salvage Cases (*n* = 65)	Overall (*n* = 240)
	HR (*p*, 95% CI)	HR (*p*, 95% CI)	HR (*p*, 95% CI)
Margin positive (ref. margin negative)	1.25 (0.46, 0.68–2.28)	1.96 (0.14, 0.78–4.90)	1.36 (0.22, 0.82–2.25)
Paraglottic space invasion	1.47 (0.07, 0.95–2.28)	1.59 (0.15, 0.83–3.03)	1.49 (0.02, 1.04–2.14)
Preepiglottic space invasion	2.23 (0.01, 1.36–3.66)	1.16 (0.75, 0.45–2.98)	1.87 (0.04, 1.21–2.88)
Lymphatic invasion	1.67 (0.03, 1.04–2.68)	1.74 (0.13, 0.84–3.60)	1.80 (0.02, 1.23–2.62)
Vascular invasion	0.87 (0.78, 0.34–2.22)	3.87 (0.07, 0.88–17.04)	1.41 (0.37, 0.65–3.04)
Perineural invasion	2.73 (0.01, 1.47–5.09)	2.80 (0.01, 1.28–6.12)	3.19 (0.01, 1.99–5.12)

ref: reference, HR: Hazard ratio, CI: Confidential interval.

**Table 4 jcm-12-00449-t004:** Risk factor analysis for 5-year disease-free survival.

Parameters	Fresh Cases (*n* = 175)	Salvage Cases (*n* = 65)	Overall (*n* = 240)
	HR (*p*, 95% CI)	HR (*p*, 95% CI)	HR (*p*, 95% CI)
Margin positive (ref. margin negative)	1.14 (0.66, 0.62–2.08)	1.96 (0.14, 0.78–4.89)	1.26 (0.35, 0.76–2.08)
Paraglottic space invasion	1.43 (0.10, 0.92–2.21)	1.51 (0.20, 0.79–2.87)	1.42 (0.05, 0.99–2.03)
Preepiglottic space invasion	2.37 (0.01, 1.45–3.88)	1.20 (0.67, 0.47–3.09)	1.92 (0.03, 1.25–2.96)
Lymphatic invasion	1.80 (0.09, 1.15–2.83)	1.84 (0.09, 0.89–3.79)	1.74 (0.04, 1.19–2.54)
Vein invasion	1.21 (0.68, 0.48–2.99)	2.89 (0.15, 0.67–12.51)	1.40 (0.38, 0.65–3.02)
Perineural invasion	2.96 (0.01, 1.62–2.96)	2.74 (0.01, 1.26–5.92)	3.05 (0.01, 1.90–4.88)

ref: reference, HR: Hazard ratio, CI: Confidential interval.

**Table 5 jcm-12-00449-t005:** Pattern of recurrence according to the status of PNI.

Parameters	PNI (+), *n* = 30	PNI (−), *n* = 210	HR (*p*, 95%CI)
Local recurrence	3 (10%)	8 (3.8%)	5.02 (0.02, 1.28–9.66)
Regional recurrence	2 (6.0%)	9 (4.29%)	1.25 (0.19, 0.15–2.36)
Distant metastasis	3 (10.0%)	19 (9.0%)	0.67 (0.23, 0.61–7.17)
Total recurrence	8 (26.7%)	36 (17.1%)	3.06 (0.04, 1.57–6.00)

PNI: Perineural invasion, HR: Hazard ratio, CI: Confidential interval.

## Data Availability

All available information is contained within the manuscript.

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
