# Peer review of "Perineural Invasion Predicts Local Recurrence and Poor Survival in Laryngeal Cancer"

_jcm, 2023, doi:10.3390/jcm12020449_

Round 1

Reviewer 1 Report

The article “Perineural invasion predicts local recurrence and poor survival in laryngeal cancer” is very interesting but I have some comments with the intention of improving the manuscript.

Material and Methods:

- Why did you consider including the smoking habit “≤, > 40 pack-years

Results:

- Line 148: “approximately” is not very appropriate word.

Discussion:

- You must include what the present study contributes in relation to previous research, for example, contrast the results with those of studies carried out by Zhu et al, Chirilá et al, Rahima et al...

- Another major limitation that should be pointed out is the lack of a multivariate analysis that included other factors predicting survival potential confounding factors, both clinical and immunological.

Conclusion:

- “an independent”, on what basis do you assert that PNI was “an independent”, significant negative prognostic marker for local recurrence after treatment termination?

Author Response

The article “Perineural invasion predicts local recurrence and poor survival in laryngeal cancer” is very interesting but I have some comments with the intention of improving the manuscript.

Material and Methods:

- Why did you consider including the smoking habit “≤, > 40 pack-years”

It was investigated as basic information since it is a factor highly associated with the occurrence of laryngeal cancer. It does not seem to have much significance for this study.

Results:

- Line 148: “approximately” is not very appropriate word.

Thank you for the kind point out. The appropriate “approximately” term was deleted.

Discussion:

- You must include what the present study contributes in relation to previous research, for example, contrast the results with those of studies carried out by Zhu et al, Chirilá et al, Rahima et al...

Thank you for your advice. We added up some comment in line 266-268

“Furthermore, the cause of treatment failure was not clear in previous studies, but we fig-ured out that PNI increased the risk of local failure and there was no significant correla-tion with regional failure and distant metastasis.”

- Another major limitation that should be pointed out is the lack of a multivariate analysis that included other factors predicting survival potential confounding factors, both clinical and immunological.

We agreed with your opinion. We added comment about that in line 264.   

Conclusion:

- “an independent”, on what basis do you assert that PNI was “an independent”, significant negative prognostic marker for local recurrence after treatment termination?

Thank you. The term “independent” seem to be inappropriate. We deleted that word.

Reviewer 2 Report

The authors analyzed the relation between clinical and pathological characteristics including perineural invasion (PNI) using the cohort of 240 laryngeal cancer patients with open surgeries. Their analyses demonstrated that PNI was an independent predictive marker of poor OS/DFS and local recurrence. Their report is really interesting and has an impact for head and neck oncologists. Some minor issues should be reconsidered.

Line 57, tonsils?  

Line 118, 73 patients (30.4%) is described as 13 (5.4%) in Table 2.

In Table 3 and 4, The reason why Margin positive are not prognostic factors of OS and DFS should be discussed.

Line 137-, paraglottic space invasion should not be significant, because its p value is 0.05 (It is incompatible with line 93 of Material and Methods).

 Line 151, p = 0.004 is shown as 0.04 in Table 5.

Author Response

The authors analyzed the relation between clinical and pathological characteristics including perineural invasion (PNI) using the cohort of 240 laryngeal cancer patients with open surgeries. Their analyses demonstrated that PNI was an independent predictive marker of poor OS/DFS and local recurrence. Their report is really interesting and has an impact for head and neck oncologists. Some minor issues should be reconsidered.

Line 57, tonsils?  

Thank you for kind point out. We corrected tonsils to larynx.

Line 118, 73 patients (30.4%) is described as 13 (5.4%) in Table 2.

Thank you. We corrected incorrect numbers according to Table 2.

In Table 3 and 4, The reason why Margin positive are not prognostic factors of OS and DFS should be discussed.

Thank you for the nice comment. We described that point in line 251-261 as below:

“In this study, the impact of positive surgical margin on 5-year OS and DFS was not statistically demonstated. In several previous studies, positive surgical margin has been known to adversely affect local control rate and survival of the disease(61, 62).Hinerman et al.(62) reported a 5-year locoregional control rate of 56% and 89% ( p = 0.075) for nega-tive and positive surgical margins in laryngeal cancer, respectively. However, the poten-tial role of surgical resection margin is still being debated, since there are no definitive da-ta available which takes into account the heterogeneity of laryngeal cancer, from a staging and biomolecular point of view(63). Furthermore, recent studies have shown that the posi-tive resection margin does not significantly affect the prognosis of laryngeal cancer pa-tients and these results are presumed to be the influence of adjuvant chemotherapy and radiation therapy(63, 64).”

 Line 137-, paraglottic space invasion should not be significant, because its p value is 0.05 (It is incompatible with line 93 of Material and Methods).

Thank you. We have checked the error and deleted “paraglottic space invasion” from that paragraph.

 Line 151, p = 0.004 is shown as 0.04 in Table 5.

Thank you. We corrected p=0.004 to 0.04 according to table 5.
